# Consumption of Galactose by *Trypanosoma cruzi* Epimastigotes Generates Resistance against Oxidative Stress

**DOI:** 10.3390/pathogens11101174

**Published:** 2022-10-11

**Authors:** Ángel Lobo-Rojas, Ender Quintero-Troconis, Rocío Rondón-Mercado, Mary Carmen Pérez-Aguilar., Juan Luis Concepción, Ana Judith Cáceres

**Affiliations:** Laboratorio de Enzimología de Parásitos, Departamento de Biología, Facultad de Ciencias, Universidad de Los Andes, Mérida 5101, Venezuela

**Keywords:** Galactose metabolism, oxidative stress, phosphoglucomutase activity, glycolysis, pentose phosphate pathway, thiol-based metabolism, hydrogen peroxide, methylene blue

## Abstract

In this study, we demonstrate that *Trypanosoma cruzi* epimastigotes previously grown in LIT medium supplemented with 20 mM galactose and exposed to sub-lethal concentrations of hydrogen peroxide (100 μM) showed two-fold and five-fold viability when compared to epimastigotes grown in LIT medium supplemented with two different glucose concentrations (20 mM and 1.5 mM), respectively. Similar results were obtained when exposing epimastigotes from all treatments to methylene blue 30 μM. Additionally, through differential centrifugation and the selective permeabilization of cellular membranes with digitonin, we found that phosphoglucomutase activity (a key enzyme in galactose metabolism) occurs predominantly within the cytosolic compartment. Furthermore, after partially permeabilizing epimastigotes with digitonin (0.025 mg × mg^−1^ of protein), intact glycosomes treated with 20 mM galactose released a higher hexose phosphate concentration to the cytosol in the form of glucose-1-phosphate, when compared to intact glycosomes treated with 20 mM glucose, which predominantly released glucose-6-phosphate. These results shine a light on *T. cruzi*’s galactose metabolism and its interplay with mechanisms that enable resistance to oxidative stress.

## 1. Introduction

Chagas disease, caused by *Trypanosoma cruzi* Chagas (Class: Kinetoplastea, Order: Trypanosomatida) affects approximately 20 million people in tropical and subtropical areas in America, and 120 million people are at risk of contracting it [1]. Although many efforts have been made to control, prevent, and cure the disease, a vaccine still does not exist, and chemotherapy based on nifurtimox or benznidazole generates significant side effects [2]. Understanding which electron-equivalent sources the parasite utilizes, and the mechanisms behind the production of molecules enhancing *T. cruzi*’s resistance against oxidative stress caused by reactive oxygen and nitrogen species (ROS; RNS), is crucial to elucidate how *T. cruzi* deals with oxidative stress caused by trypanocidal drugs and host immune defenses, since both nifurtimox and benznidazole seem to be able to produce an imbalance in redox homeostasis, which is extremely effective at promoting parasite lysis [3,4].

During its complex life cycle, *T. cruzi* is exposed to damaging reactive oxygen species (ROS) [5,6,7,8] from its internal metabolism, as well as to high concentrations of ROS and reactive nitrogen species (RNS) generated by cytotoxic reactions mediated by the host’s immune system [9,10]. Furthermore, redox-active thiol groups in proteins and low molecular mass compounds play key roles as buffers that balance any disturbance of the intracellular redox state [11].

In Trypanosomatids, the most important intracellular redox buffer is trypanothione (N1,N8-bisglutathionylspermidine) [12], which is a compound with low molecular mass synthesized by trypanothione synthetase in an ATP-dependent reaction in which two molecules of glutathione are covalently linked by a spermidine molecule [13]. Trypanothione is found in two forms: a reduced form called dihydrotrypanothione (T[SH]_2_), which functions as electron donor, and an oxidized form, trypanothione disulfide (TS_2_). The electron acceptors of trypanothione are a battery of low-molecular-mass thiol disulfide oxidoreductase enzymes (which include tryparedoxin peroxidase, a 2-Cys peroxiredoxin that directly reduces peroxides [14]), and a cytosolic non-selenium GSH-peroxidase-like tryparedoxin peroxidase. Both enzymes act as the main peroxide detoxifiers in the cytosol of these parasites [15,16,17], and belong to an intricate network converging to T[SH]_2_. Trypanothione is recycled to its reduced form by the activity of the cytosolic NADPH-dependent trypanothione reductase (TR) [18,19]. The reaction catalyzed by this enzyme represents the only known connection between NADPH-producing reactions and thiol-based redox metabolisms, since glutathione reductases and thioredoxin reductases have not been observed in Trypanosomatids [15,20,21].

Additionally, the fact that reactions catalyzed by the glucose-6-phosphate dehydrogenases (G6PDH) and 6-phosphogluconate dehydrogenases (6PGDH) from the pentose phosphate pathway (PPP) are localized both in the cytosol and glycosomes [22], and the reaction catalyzed by the malic enzyme (ME) contributes with the cytosolic NADPH pool when parasites are glucose starved [23,24,25], help Trypanosomatids to maintain a suitable cytosolic NADPH:NADP^+^ ratio.

Experiments demonstrate that glucose metabolism fluxes shift towards the PPP when *Leishmania donovani* is exposed to oxidative stress [24]. Glucose is the main hexose feeding the PPP in Trypanosomatids via the production of glucose-6-phosphate (G6P) by hexokinase (HK). However, this is not the only source of this sugar–phosphate. Recently, the Isselbacher pathway has been described for both *Leishmania major* and *T. cruzi* [26,27,28,29,30,31,32,33]. Galactose can also be used to produce UDP-glucose and UDP-galactose, which are the main substrates for all glycosylation routes, and for G6P production. This pathway is composed by the enzymes galactokinase (GALK), UDP-sugar pyrophosphorylase, and UDP-galactose 4′-epimerase, which yield glucose-1-phosphate (G1P) from α-D-galactose. Therefore, phosphoglucomutase (PGM) uses G1P to produce G6P, which has two possible outcomes: glycolysis and the PPP [33].

Although epimastigotes of *T. cruzi* can consume galactose, parasites that grow in the presence of this sugar produce higher ammonium concentrations when compared to parasites growing in presence of glucose, which suggests that galactose is metabolized differently to glucose. However, the underlying mechanism remains unclear [33].

In this study, we uncovered a relationship between exogenous galactose consumption and an increase of *T. cruzi* epimastigote resistance against oxidative stress caused by hydrogen peroxide and methylene blue (MB).

## 2. Results

### 2.1. Epimastigotes Supplemented with Galactose Showed Higher Resistance against Oxidative Stress Produced by Exposure to Hydrogen Peroxide than Those Supplemented with Glucose

Epimastigotes were grown in 20 mM glucose, 1.5 mM glucose (which is consumed within two days of growth, after which the culture starts consuming amino acids—see Appendix A), and 20 mM galactose. These initial growth conditions allowed the parasites to adapt their metabolisms to their available primary carbon and energy sources (glucose, galactose, or amino acids). Next, the parasites were transferred to the oxidative stress buffer (OSB), and the appropriate volume of hydrogen peroxide was added in order to reach the desired final concentration. As shown in Figure 1 (panels A, B, and C), the parasites whose metabolisms were primed with galactose and maintained in presence of the appropriate hexose had a higher percentage of viability when compared to the other two growth conditions (20 mM glucose, and 1.5 mM glucose).

In the 50 μM hydrogen peroxide assay, the effect of exposition begins to be evident just after initiating the experiment, at 15 min of exposition. Parasites grown in 20 mM galactose displayed a higher viability percentage when compared to the other two treatments (20 mM galactose assay: 85 ± 2%; 20 mM glucose assay: 59 ± 2%; 1.5 mM glucose assay: 40 ± 2%).

*T. cruzi* epimastigotes grown in 20 mM LIT galactose and exposed to a sub-lethal concentration of hydrogen peroxide (100 μM) [34] were 2.3 times more resistant than those grown in 20 mM LIT glucose (67 ± 2% vs. 29 ± 2% of viable cells, respectively), and 5.2 times more than the parasites grown in 1.5 mM LIT glucose (67 ± 2% vs. 13 ± 1.1% of viable cells, respectively). It is worth noting that, after the initial exposure of the epimastigotes to hydrogen peroxide, a rapid decline in viable cell numbers can be observed in both assays, except for the parasites that were supplemented with galactose (Figure 1, panels B and C).

A similar result was obtained when calculating IC_50_ values. When the parasites were supplemented with galactose, the epimastigotes exhibited an IC_50_ value 2.35-fold and 4.30-fold higher when compared to the epimastigotes grown in standard LIT (20 mM glucose), and those grown in low-glucose LIT media (1.5 mM glucose), respectively (see Appendix A).

Decomposition rates of hydrogen peroxide were constant across all assays. This may have been caused by the plethora of possible reactions that can occur due to ROS interacting with lipids, nucleic acids, and proteins [5,6,7,8]. Hydrogen peroxide decays exponentially in a concentration-specific fashion (see Appendix A).

The higher observed viability or resistance of the epimastigotes grown in the presence of 20 mM galactose and exposed to hydrogen peroxide may be due to three possible causes: (1) the epimastigotes may be producing an enriched glycocalyx as a consequence of the production of a higher concentration of UDP-sugars, which are a possible byproduct of galactose metabolism, thus preventing the inflow of hydrogen peroxide and derived molecules into the cells; (2) the expression of peroxide detoxifying systems, such as glutathione-peroxidase-type peroxidases (GPX’s), peroxiredoxin-type peroxidases (TXNPx’s), ascorbic peroxidase (APx) and/or a higher production of trypanothione and other low-mass thiol molecules may be impacted in epimastigotes grown in the presence of different carbon sources (glucose, galactose, and amino acids). This may contribute to the observed higher resistance to oxidative stress, and/or (3) the parasites may increase their reductive capabilities by producing NAPDH (produced during the PPP), which in turn may augment the T[SH]_2_/TS_2_ and other thiol-related molecule ratios that could scavenge the main parts of the derived radicals from hydrogen peroxide, possibly explaining the higher observed viability of the epimastigotes grown with galactose as the main carbon source.

### 2.2. Epimastigotes Supplemented with Galactose Showed Higher Resistance against Oxidative Stress Produced by Methylene Blue as Compared to Those Grown with Glucose and Those Grown in the Absence of Glucose

In order to test the aforementioned hypotheses, we carried out an additional assay using MB, which a) acts as a di (thiol)-oxidizing agent, b) serves as a subversive substrate of the flavoenzymes TR and dihydrolipoamide dehydrogenase [MB^+^ + NAD(P)H → leucoMB + NAD(P)^+^]; the nascent colorless leucomethylene blue (leucoMB) is instantaneously auto-oxidized, yielding MB and hydrogen peroxide, and c) non-competitively inhibits the TS_2_ reduction by TR (K*_i_* = 1.9 µM) [35], thus acting as a more effective stressor when compared to hydrogen peroxide.

Although methylene blue can impact redox metabolism at different levels, our attention was drawn to the fact that MB can be used as a subversive substrate, enabling the production of hydrogen peroxide using NAD(P)H as an electron source, which is why we chose to use 30 µM of MB, which has been previously reported as the K*_m_* for TR [35].

Figure 2B shows that parasites supplemented with galactose had a higher viability (70%) after 120 min of incubation with MB when compared to parasites grown in the presence of 20 mM glucose and 1.5 mM glucose using the same incubation time, for which the viability was notably lower (45% and 37%, respectively), being statistically significant (see Appendix A). In Appendix A, a strong decline in the MB concentration is observed for parasites supplemented with 20 mM galactose between 0 and 5 min. After this first phase, the MB concentration remains relatively constant. Approximately 10 μM of MB were reduced by the parasites supplemented with galactose; in contrast, only 1 and 2 μM were reduced by the parasites grown under the remaining two conditions. As observed by Rengelshausen et al. [36], there is a rapid equilibrium between leucoMB and MB in vivo; this depends mainly on dithiol molecules, which are capable of reducing MB, and enzymes such as TR that reduce MB using NADPH as an electron donor. Additionally, in vitro studies carried out by Buchholz et al. [35] demonstrated that NADPH and MB reached rapid equilibrium in the presence of TR, which remained stable even under aerobic conditions.

Our results (see Figure 2 and Appendix A) suggest that increased NADPH availability on epimastigotes supplemented with galactose is the most likely mechanism behind enhanced parasite viability, as there is no known reaction between MB and the glycocalyx, glycoconjugates, or the major peroxide detoxifying enzyme systems; however, several reactions between thiol molecules and MB have been studied [35,37].

### 2.3. Phosphoglucomutase Activity Is Localized in Both Glycosomes and the Cytosol, but Is Mainly Cytosolic

In order to understand why galactose metabolism induces higher resistance against oxidative stress, we studied the subcellular localization of PGM activity, which acts as the link between the Isselbacher pathway and glucose metabolism (glycolysis/PPP). All Isselbacher pathway enzymes are localized within glycosomes [33,38,39], as they are the first six or seven glycolytic enzymes [40]; the PPP is located in both the glycosome (30%) and the cytosol (70%) [22], while TR and trypanothione-dependent redox systems are also cytosolic. Thus, the subcellular localization of PGM activity is important to understanding the higher observed resistance against oxidative stress induced by galactose consumption. Although the subcellular localization of the PGM protein was discovered by Penha et al. [26] using immunofluorescence and immunocytochemistry techniques, the subcellular distribution of PGM activity has never been determined in the parasite.

Therefore, we used two methods to study the subcellular localization of PGM activity in epimastigotes of *T. cruzi*: differential centrifugation, and partial permeabilization with digitonin. PGM activity was enriched in the cytosolic fraction (Figure 3A); the distribution was similar to that determined for pyruvate kinase (PyK) (Figure 3B), which is a classical cytosolic marker in Trypanosomatids [41]. However, PGM activity was not enriched in the glycosome-rich fraction (SG); in this regard, it is unlike the HK (glycosomal marker enzyme) [42], phosphoglucose isomerase (PGI), which is located in both the cytosol (C) and the glycosome (SG) [43], and malate dehydrogenase (MDH), whose activity shows enrichment in large and small granular fractions (Figure 3C–E, respectively) [44].

Figure 3F shows that the liberation profile of PGM activity was enriched in assays with low digitonin concentrations; this is similar to the cytosolic marker enolase (ENO), which was completely released with about 0.08 mg digitonin × mg^−1^ protein, and 60% of PGI activity was liberated with the same concentration of digitonin, corresponding with the cytosolic isoenzyme [44]. HK activity and 40% of PGI activity (glycosome isoenzyme) were released at 0.16 mg digitonin × mg^−1^ protein, corresponding with typical glycosome enzymes. The release of the main part of PGM activity (~90%) was similar to that of ENO, and just ~10% was liberated completely in 0.16 mg digitonin × mg^−1^ protein. The obtained biphasic curve is similar to that of PGI activity, which implies that PGM activity may be localized in both glycosomes and the cytosol, but is predominantly cytosolic. This corroborates previous results in which the PGM protein was detected in both compartments [26].

### 2.4. Determination of the Generation Time, Specific Activities of Some Enzymes of the Intermediary Metabolism, and the Consumption Rates of Glucose, Galactose, and Amino Acids in Epimastigotes Grown in the Presence of These Three Carbon Sources

Table 1 shows the specific activities of several enzymes measured in homogenates of epimastigotes grown in the presence of different carbon sources. HK (the glycolytic marker) showed the same order of magnitude in all treatments (independent of the carbon source). Furthermore, it is consistent with specific activities previously reported for *T. cruzi* homogenates [42] grown in a standard LIT medium (20 mM glucose). NAD(H)-dependent L-glutamate dehydrogenase (GlDH-NAD^+^, amino acid degradation marker) also shows a specific activity that is consistent with previous results obtained for epimastigotes grown in a standard LIT medium [45,46,47]. GALK also shows consistency with previous results, whose specific activity corresponds to both *Trypanosoma cruzi* GALKs 1 and 2 [33]. G6PDH, the first enzyme of the PPP, also shows specific activity consistent with previous works [48,49]. No reports about PGM’s specific activity in epimastigotes, or any other stage of the parasite, have been published.

However, the specific activities of GALK and PGM increased significantly, (four-fold and two-fold, respectively) in parasites supplemented with galactose, compared to the other two assays. Similarly, GlDH-NAD^+^-specific activity in parasites grown in a galactose-supplemented LIT medium increased 3-fold when compared to parasites grown in a low-glucose LIT medium, and 1.25-fold when compared to parasites grown in a high-glucose LIT medium; this agrees with previous results, where parasites grown in a LIT medium supplemented with galactose consume galactose and amino acids concomitantly [33].

Table 2 shows the measured parameters that may be useful for numerically comparing the usage forms of glucose (which were obtained in the exponential phase, as shown in Appendix A), galactose, and amino acids for *T. cruzi* epimastigotes. Parasites supplemented with 20 mM galactose showed a similar generation time (G) when compared to parasites supplemented with 20 mM glucose, while parasites not supplemented with hexoses showed a G approximately three times lower. Galactose consumption rates are approximately half that of glucose. However, it is important to note that amino acid consumption, represented by the excretion of ammonium, is three times higher in parasites supplemented with galactose, when compared to parasites grown in high-glucose LIT medium. These excretion rates are 1.5 times higher than those shown by parasites lacking any hexose as a carbon source (low-glucose LIT medium).

These galactose consumption rates may be too high to be sustained only by the PGM-specific activity found in glycosomes, which corresponds to ~10% of the total specific activity, which suggests that one or more unknown mechanisms diverting part of the G1P produced inside the glycosomes into the cytosol may be at play. Nevertheless, we cannot exclude the possibility that an unknown activator inside glycosomes may be augmenting specific activity of glycosomal PGM, thus providing catalyst capability in order to process all/most of the G1P inside glycosomes.

### 2.5. Epimastigotes That Have Been Partially Permeabilized with Digitonin Consume Galactose and Mainly Produce Glucose-1-Phosphate

Taking into account previous results of specific activities, and assuming that the specific activity measured in this work is near in vivo conditions, some mechanisms of diverting glycosomal G1P to cytosol compartment may be working to metabolize all the G1P produced inside glycosomes. We thus studied one of the possible mechanisms by which hexose phosphates originated from galactose metabolism may be being transported to the cytosol, using digitonin (30 µg × mg^−1^ of protein for 20 min) to partially permeabilize epimastigotes grown in a standard LIT medium (20 mM glucose), which were then harvested during the exponential growth phase. Additionally, we supplemented the resuspended parasites with phosphoenolpyruvate (PEP), ADP, NaHCO_3_, and L-malate to restore the glycolytic flux in partially permeabilized epimastigotes, as reported by Sanz-Rodríguez et al. [50]. We then studied the inorganic phosphate-dependent transportation of G1P and G6P to the supernatant over the following two hours.

Figure 4 shows the consumption rates of glucose and galactose, and the concomitant release of G6P and G1P. It is worth noting that G6P is mainly released to the supernatant when partially permeabilized epimastigotes consume glucose; in contrast, it is mainly G1P that is released to the supernatant when the partially permeabilized epimastigotes consume galactose. This result agrees with the observed subcellular localization of PGM activity, which is mainly cytosolic and essential for G1P and G6P interconversion. Under our experimental conditions, neither G1P nor G6P are released to the supernatant when partially permeabilized epimastigotes are supplemented with glucose or galactose in the absence of inorganic phosphate (see Figure 4B,D). These observations suggest the existence of an anti-porting mechanism that allows for the inorganic phosphate-dependent release of hexose phosphates to the cytosol, as observed in plant plastids [51,52,53,54,55] and in the human endoplasmic reticulum [56,57].

## 3. Discussion

Our results suggest that epimastigotes grown using galactose as the main carbon source are more resistant to oxidative stress than epimastigotes grown either in the presence or absence of glucose, since they display higher viability percentages after being submitted to oxidative stress by exposure to hydrogen peroxide or MB. Furthermore, the IC_50_ values calculated for hydrogen peroxide were shown to be higher in epimastigotes supplemented with galactose, than in epimastigotes supplemented with glucose.

Given that both hexoses’ oxidation to CO_2_ and H_2_O contain the same Gibbs free energy, if galactose were to be completely metabolized to G6P inside glycosomes, both hexoses might be equivalent from the metabolic point of view in *T. cruzi*. However, our results suggest that each hexose feeds glycolysis and the PPP in different proportions.

There are important differences between the effects caused by hydrogen peroxide and those caused by MB. Hydrogen peroxide has nonspecific effects, via the production of derived ROS that may affect the functionality of almost any biomolecule [5,6,7,8]. For this reason, it is not possible to determine whether the resistance against hydrogen peroxide shown by parasites supplemented with galactose is due to a higher redox buffering capacity resulting from the NADPH produced on the PPP, or if it is due to a significant difference regarding the quantity and types of biomolecules that make up the parasites themselves. For instance, a higher concentration of glycoconjugates (whose production may be enhanced as a result of galactose metabolism and sugar–nucleotide synthesis) could limit the import of hydrogen peroxide into cells, as derived ROS might react with these kinds of molecules in the exterior of the parasite cells. Additionally, we cannot exclude the possibility that the galactose metabolism could be up-regulating (by an unknown mechanism) the activity of some peroxide-detoxifying enzymes such as ascorbate peroxidase [58], tryparedoxin peroxidase [14], cytosolic and mitochondrial GSH-peroxidase-like tryparedoxin peroxidase [59], which can detoxify the hydrogen peroxide directly.

On the other hand, MB has a specific oxidant effect on several redox buffer thiol molecules, such as glutathione, glutathionylspermidine, trypanothione, dihydrolipoamide, etc. Additionally, it is important to note that MB serves as a subversive substrate of TR, and dihydrolipoamide dehydrogenase, oxidizing NADPH in the enzymatic process [35]. Therefore, the higher resistance to oxidative stress caused by exposure to MB shown by galactose-supplemented epimastigotes suggests that the consumption of this hexose increases the amount of reduced thiol molecules as the most likely result of a high cytosolic ratio of NADPH:NADP^+^; this conclusion is supported by the fact that galactose-consuming epimastigotes exported a higher concentration of hexose phosphates in the form of G1P to the cytosol, when compared to their glucose-consuming counterparts, which mainly exported G6P (see Figure 5). It is worth noting that PPP is the only known pathway in Trypanosomatids utilizing the G6P being exported to the cytosol. Additionally, a rapid reduction in the MB concentration is observed in epimastigotes supplemented with galactose when compared to the other two treatments, which reflects their higher capacity to buffer redox changes.

Galactose is metabolized through the Isselbacher pathway to produce G1P and other intermediates that may be used as precursors for protein glycosylation [33,60]. Likewise, PGM catalyzes the reversible reaction between G6P and G1P, connecting the Isselbacher pathway to glycolysis and PPP. Penha and colleagues [26] demonstrated, via immunofluorescence assays, that PGM displays a dual localization pattern: a diffused cytosolic pattern and a punctuated glycosomal pattern. We also found a dual localization of PGM activity in *T. cruzi* epimastigotes, with ~90% of the PGM activity being localized in the cytosol, and the remaining ~10% being localized in the glycosome. However, in proteomic studies conducted on purified glycosomes obtained from *T. cruzi* epimastigotes, PGM was not detected [61], while the presence of a phosphomannomutase (PMM) with a signal sequence PTS-1 (-SNL) was discovered; this was recently corroborated as a true PMM by Zmuda et al. [62]. Until now, it has not been determined whether *T. cruzi*’s PMM has PGM activity: this is a plausible possibility, as it is reported to occur in *T. brucei*. PMM and PGAM (phospho-N-acetylglucosamine mutase), in addition to their canonical activity, substitute PGM activity in this related organism [63]; therefore, we do not rule out the possibility that, in *T. cruzi*, PGM activity associated with glycosomes could be the product of PMM and not of true PGM.

PGM-specific activity inside glycosomes could be too low to sustain galactose consumption, as can be inferred by comparing it with galactose consumption rates (approximately 0.7 nmol × min^−1^ × mg^−1^ inside glycosomes, vs. 2.74 nmol × min^−1^ × mg^−1^ of total galactose consumption). This would constitute a bottleneck in galactose metabolism, which raises the idea that a higher proportion of G1P is exported to cytosol (see Figure 5). In fact, we have demonstrated the existence of anti-porting mechanisms of G1P and G6P, using inorganic phosphate as a counter-ion, although we cannot rule out the possibility that other mechanisms that are diverting hexose phosphates may be operating. G1P transported to cytosol can be transformed into G6P via PGM activity, thus feeding the 70% of the PPP that is localized in this compartment [22], leading to NADPH production via the oxidative branch of the PPP (see Figure 5). However, part of the G1P produced inside glycosomes may be metabolized through glycolysis and glycosomal PPP, as a non-negligible amount of PGM-specific activity occurs inside glycosomes.

Both the assays that used parasites supplemented with galactose and those that used parasites without supplemented glucose displayed higher rates of ammonium excretion when compared to parasites supplemented with glucose. The consumption of energy-rich molecules is accompanied by changes in the specific activities of several enzymes, which reflect the adaptability of the parasite’s metabolism to the new carbon source. In the presence of galactose, GALK-specific activity increases in the same order of magnitude as do PGM and GlDH-NAD^+^, while the specific activity of HK and G6PDH remain in the same order of magnitude regardless of the energy and carbon source used. The observed increase in amino acid metabolism in epimastigotes supplemented with galactose could be a consequence of the need for ATP to feed the cytosolic biosynthetic pathways. Galactose metabolism may yield a lower ATP synthesis than glucose does. When we fed galactose to partially permeabilized epimastigotes, a higher proportion of G1P was released from intact glycosomes than from epimastigotes supplemented with glucose (see Figure 4). Under these conditions, galactose may be a primary source of NADPH (see Figure 5), ribose, and 5-phosphoribosyl-1-pyrophosphate (for nucleotide biosynthesis) through the PPP. However, it should be noted that other cytosolic dehydrogenases, such as NADP^+^-dependent isocitrate dehydrogenases, glutamate dehydrogenases, and ME, may also provide additional NADPH in the cytosol during amino acid consumption, as observed in *T. brucei* [23] (see Figure 5).

Cytosolic NADPH is an essential electron source for the detoxification of radical species; it is achieved through the action of a battery of enzymes (for more information, see [20,64,65,66,67]) which use T[SH]_2_ as an electron source. Ultimately, the reduction of TS_2_ to T[SH]_2_ occurs exclusively through the action of TR, which uses NADPH as an electron donor. It is important to note that all radical species’ detoxifying reactions always converge on their electron source coming from T[SH]_2_, and ultimately from NADPH.

*Trypanosoma cruzi* can encounter significant amounts of galactose during its life cycle, since the calculated concentration of galactose in human adults’ blood-streams is around 1 mM [68]. In the epimastigote form, parasites would be able to take advantage of the galactose arriving to the midgut every time the vector takes a fresh blood meal. Meanwhile, in the blood-stream trypomastigote form, the parasite would constantly be in the presence of the hexose. On the other hand, while in the intracellular stages (trypomastigotes and amastigotes), galactose uptake by the parasite may potentially compete with human cytosolic GALK for the substrate.

The initial stage of *T. cruzi* infection is maintained under host control by processes such as phagocytosis and nitric oxide production by cells such as macrophages, T lymphocytes, antigen-presenting cells, neutrophils, and natural killer cells [69,70]. After phagocytosis, and inside the phagolysosomes, trypomastigotes are submitted to a highly oxidative environment as many mechanisms for the production of ROS and RNS are activated by the host (e.g., NADPH oxidase, which produces superoxide radicals [5,71]; cytoplasmic nitric oxide synthase, which produces nitric oxide that diffuses to the phagolysosomes and reacts with superoxide radicals to form many free radicals; and, most importantly, peroxynitrite production, which acts as a potent oxidant against *T. cruzi* [72,73]). High concentrations of free radicals inside the phagolysosome are detrimental for the parasite [10]. Therefore, in order to survive and support trypomastigote migration into the cytoplasm, *T. cruzi* must effectively cope with oxidative stress; NADPH production derived from galactose metabolism may play an important role here.

Once inside the cytoplasm, the amastigote stage is exposed to lower concentrations of ROS and RNS [74]. The innate immune response has shown to be critical for preventing a high parasite burden in the acute stage, which leads to the development of other stages of Chagas disease, and improving the prognosis of the individuals. On the other hand, if high parasite proliferation occurs in the acute stage as a consequence of improved oxidative burst management for *T. cruzi*, the prognosis for human individuals is mostly poor and hopeless [75,76,77].

As such, galactose metabolism may play an essential role in the parasites’ ability to deal with oxidative stress bursts during the initial stages of the infection. The manner in which galactose is metabolized may be considered a significant pathway leading to oxidative stress management, and, consequently, may be seen as a step towards chemotherapy using nifurtimox and benznidazole due to their modes of action [78].

It could be worth studying the relationship between galactose consumption and resistance against trypanocidal drugs, as well as testing specific inhibitors that exhibit downstream effects on the galactose metabolism, such as G6PDH inhibitors [79,80] or TR specific inhibitors [81]. This may help to elucidate key steps of the resistance against oxidative stress observed in galactose-consuming parasites.

## 4. Materials and Methods

### 4.1. Parasites, Culture Media, and Growth Conditions

*T. cruzi* strain EP epimastigotes were cultured axenically at 28 °C, with constant shaking at 160 RPM, in a liver infusion-tryptose (LIT) medium [82], supplemented with 5% heat-inactivated fetal bovine serum.

All parasites used in our experiments were grown as described by Lobo-Rojas et al. [33]: (i) a low-glucose, partially depleted medium (α-D-glucose 1.5 mM, supplemented with 5% heat-inactivated fetal bovine serum dialyzed by tangential flow filtration against NaCl 0.15 M (Life Technologies, Carlsbad, CA, USA, D-glucose concentration < 0.0278 μM)). When required, this medium was supplemented with either (ii) 20 mM α-D-glucose (high-glucose) or (iii) 20 mM α-D-galactose 20 mM (Sigma-Aldrich (St. Louis, MO, USA) ≥ 99%).

### 4.2. Determination of the Generation Time, and Measurement of Hexose Consumption and Ammonium Excretion Rates

The epimastigote generation time (G) was calculated as described by González-Chávez et al. [83]. The measurement of the specific hexose consumption and ammonium excretion rates (q) was made according to Stouthamer et al. [84] and Haanstra et al. [85] as follows:$$q_{Metabolite}=\frac{\mu }{Y_{metabolite}}$$
where *μ* is the growth rate constant during the exponential phase and *Y_metabolite* is the absolute value of the slope of the cell density curve plotted against the metabolite concentration during the exponential growth phase.

### 4.3. Determination of the Oxidative Stress Resistance of Epimastigotes Grown in Different Carbon Sources Using Hydrogen Peroxide and Methylene Blue as Oxidant Agents

Parasites were harvested in the middle of the exponential growth phase (see Appendix A) and washed three times with an oxidative stress buffer (OSB), which had the following composition: 16.2 mM Na_2_HPO_4_, 3.8 mM NaH_2_PO_4_, 5 mM KCl, 95 mM NaCl, 2 mM MgCl_2_, pH 7.4, supplemented with 20 mM of glucose or galactose, accordingly. A control assay was made by substituting hexoses on the low-glucose LIT-medium assay with 10 mM NaCl, to maintain the osmolarity of the buffer. The number of parasites per milliliter was counted using a Neubauer chamber, after which the cell concentration was adjusted to 1 × 10^7^ cells × mL^−1^. Epimastigotes from each of the 3 assays were then resuspended in separate Erlenmeyer flasks to a final volume of 100 mL, and a final concentration of 1 × 10^7^ cells × mL^−1^.

Then, an appropriate amount of the stock solution to reach the correct final concentrations of hydrogen peroxide was added (0, 50 or 100 μM; Finzi et al. [34]). These concentrations were subsequently quantified as described in Section 4.5. Parasites were incubated at 28 °C and 160 RPM, and 1-mL fractions were taken at 0, 15-, 30-, 60-, and 120-min intervals, after the addition of hydrogen peroxide, and centrifuged at 12,000× *g* per 2 min. Epimastigotes were then washed three times with the appropriate OSB, and parasite pellets from each assay were resuspended in the respective OSB. The percentage of viable cells was estimated using the Trypan blue exclusion test. Parasites from each assay were supplemented accordingly with their respective hexose during all experimental steps.

The effect of hydrogen peroxide on the viability of epimastigotes (IC_50_) was calculated by incubating 1 × 10^7^ parasites × mL^−1^ in OSB for 2 h, supplemented with the appropriate carbon source and in the presence of diverse concentrations of hydrogen peroxide (from 0 to 200 μM). Then, the epimastigotes were washed three times with OSB and counted using a Neubauer chamber, and dyed with Trypan blue to estimate parasite viability (see next section). Data were plotted using GraphPad Prism 8 software (see Appendix A).

Parasite resistance against oxidative stress caused by MB was assessed as follows: epimastigotes from each assay were harvested during the mid-phase exponential growth phase, washed with OSB, and resuspended, with their concentration adjusted to 1 × 10^7^ cells × mL^−1^.

The resuspended parasites were then separated into two Erlenmeyer flasks and MB was added until the appropriate concentration (0 and 30 μM) was reached. The parasites were incubated at 28 °C and 160 RPM, and 1-mL fractions were taken at 0, 5, 15-, 30-, 60-, and 120-min intervals.

Epimastigotes were then washed three times with the appropriate OSB, and pellets from each assay were resuspended accordingly in their respective OSB. The percentages of viable cells were estimated by using the Trypan blue exclusion test. Parasites from each assay were supplemented accordingly with their respective hexose during all experimental steps. Statistical analysis of two-tailed, unpaired samples t-test were performed using GraphPad Prism 8 software (see Appendix A).

### 4.4. Counting Viable Cells Using the Trypan Blue Exclusion Method

The Trypan blue exclusion test helps distinguish living cells from dead cells, as dead cells lose the ability to excrete the Trypan blue dye, thus remaining stained after the procedure, while living cells do not. A 50 μL fraction of the parasites was extracted from each assay, and was mixed with 50 μL of 0.4% Trypan blue in 0.85% NaCl. After homogenizing and incubating the suspension for 10 min, both living and dead parasites were counted using a Neubauer chamber and the percentage of viability (% Viability) was calculated as the ratio of viable cells to total cells, multiplied by 100.

### 4.5. Determination of Hydrogen Peroxide, Methylene Blue, Hexoses, Hexose Phosphates, and Ammonium Concentrations

Hydrogen peroxide concentrations were measured on the stock solution (Sigma) and the OSB, using an enzymatic assay [86] with horseradish peroxidase C type VI (Sigma), and 3,3′,5,5′-Tetramethylbenzidine (TMB) (DIAGEN CA), and by estimating the ΔOD_653nm_ and using the Lambert-Beer equation. Meanwhile, the concentration of MB was spectrophotometrically estimated using the Lambert-Beer equation, as described by Buchholz et al. [35].

Glucose, galactose, and ammonium concentrations were determined as previously described [33]. Similarly, G1P, and G6P concentrations were calculated by standard enzymatic methods. G1P was determined using 1 U × mL^−1^ of phosphoglucomutase (PGM) (from rabbit muscle, Sigma), 1 U × mL^−1^ of glucose-6-phosphate dehydrogenase (G6PDH) (type XXIV from *Leuconostoc mesenteroides*, Sigma), and 50 µM glucose-1,6-biphosphate, 0.72 mM NADP^+^. The ΔOD_340nm_ was used to calculate the G1P concentration using the Lambert-Beer equation and the extinction coefficient of NADPH at 340 nm of 6220 M^−1^ × cm^−1^. Meanwhile, the G6P concentration was calculated using the same protocol, but excluding PGM and glucose-1,6-biphosphate from the assays.

### 4.6. Determination of Hexose Consumption and Hexose Phosphate Production in Intact Glycosomes from Epimastigotes Partially Permeabilized with Digitonin

Parasites were harvested during the exponential growth phase, and processed as described by Sanz-Rodríguez et al. [50] with some modifications: the epimastigotes were collected by centrifugation at 4000× *g* for 10 min, resuspended in buffer A (75 mM Tris-HCl, 140 mM NaCl and 100 mM KCl, pH 7.4) at a final concentration of 8 × 10^8^ cells × mL^−1^, and incubated with digitonin at a concentration of 30 µg × mg^−1^ of protein for 20 min. After digitonin treatment, parasites were washed twice with buffer A, resuspended in the same buffer, and supplemented with 6 mM PEP, 6 mM ADP, 7 mM NaHCO_3_, and 1 mM L-malate.

The partially permeabilized epimastigotes were separated into two tubes: the first was supplemented with 20 mM glucose, while the second was supplemented with 20 mM galactose. Two fractions from each tube were extracted: one with added 6 mM Na_2_HPO_4_, and a control fraction with no added inorganic phosphate, for a total of 4 treatments.

The assay was started by adding either 20 mM glucose or 20 mM galactose, accordingly, and all groups were incubated at 28 °C over the course of 2 h with mild agitation (120 RPM). Then, 2 mL samples were picked up every 20 min and centrifuged at 4 °C and 12,000× *g* for 5 min to eliminate partially permeabilized epimastigotes. The supernatants were used to determine glucose, galactose, G1P, and G6P concentrations, as described in the previous section.

### 4.7. Enzymatic Assays and the Measurement of Enzyme-Specific Activities

All enzymatic assays were carried out using a Hewlett Packard (Palo Alto, CA, USA) 8452 diode array spectrophotometer at 340 nm at room temperature. PGM (EC 5.4.2.2) activity was spectrophotometrically measured at 340 nm by coupling the reaction to G6PDH (NADPH production). The assay was performed in a 1-mL cuvette containing 0.1 M Tris-HCl, pH 7.6, 50 mM KCl, 10 mM MgCl_2_, 1 mM of G1P, 50 µM glucose-1,6-biphosphate, 0.72 mM NADP^+^, and 1 U × mL^−1^ of G6PDH (type XXIV from *Leuconostoc mesenteroides*, Sigma). Other enzymes were assayed as described previously: ENO (EC 4.2.1.11) [87], HK (EC 2.7.1.1) [42], G6PDH (EC 1.1.1.49) [88], PGI (EC 5.3.1.9) [43], GALK (EC 2.7.1.6) [33], PyK (EC 2.7.1.40), GlDH-NAD^+^ (EC 1.1.1.3), and MDH (EC 1.1.1.37) as described by Bergmeyer [86]. One unit of an enzyme is defined as the amount of the enzyme that catalyzes the formation of 1 μmol of product or the conversion of 1 μmol of substrate per minute at 25 °C.

Enzyme-specific activity was determined using parasites harvested during the exponential growth phase of each assay (see Appendix A), centrifuged at 5000× *g* for 5 min at room temperature, and resuspended in a lysis buffer (70 mM Tris-HCl pH 7.4, 140 mM NaCl, 400 mM KCl, 0.2% Triton X-100, plus the protease inhibitors cocktail as described by Cáceres et al. [42]). Epimastigotes were then incubated at 4 °C for 30 min, to promote cell lysis, and again centrifuged at 12,000× *g* and 4 °C for 5 min. Enzymatic activities were measured in the supernatants; HK, GALK, G6PDH, PGM, and GlDH-NAD^+^ were assayed as described above. Protein concentration was measured using the Lowry method [89], using bovine serum albumin as a standard.

### 4.8. Subcellular Localization of Phosphoglucomutase Activity

The subcellular localization of PGM activity in *T. cruzi* epimastigotes was determined by differential centrifugation and selective permeabilization with digitonin. For the differential centrifugation step, epimastigotes (2 g wet weight) were homogenized by grinding with silicon carbide (200 mesh) and fractionated by differential centrifugation, as previously described for *T. cruzi* by Concepción et al. [90]. The fractions obtained were: nuclear (N), large granular (LG), small granular (SG; enriched in glycosomes), microsomal (M), and cytosolic (C). Specific activities from PyK (cytosolic), HK (glycosomal), PGI (cytosolic and glycosomal), and MDH (mitochondrial), were used as enzyme markers for each different fraction. Selective digitonin permeabilization was performed as described by Acosta et al. [44]. The marker enzymes used were ENO (cytosolic), HK (glycosomal), and PGI (cytosolic and glycosomal).

## 5. Conclusions

In this work, we presented results that demonstrate that *T. cruzi* epimastigotes’ usage of galactose as a carbon source provides them with significantly higher resistance against oxidative stress derived from exposure to hydrogen peroxide and MB, as compared to epimastigotes that only use glucose or amino acids as carbon sources. We also demonstrated that ~90% of PGM activity in *T. cruzi* epimastigotes is primarily localized within the cytosolic compartment. Additionally, when compared with epimastigotes supplemented with either glucose or amino acids, epimastigotes supplemented with galactose showed enhanced amino acid consumption rates, and adapted their metabolism to use galactose as a primary carbon and energy source, as suggested by changes in the specific activity of some marker enzymes.

Additionally, we found that epimastigotes that have been partially permeabilized using digitonin are able to transport G1P or G6P in an inorganic phosphate-dependent manner when supplemented with either galactose or glucose, respectively. Higher hexose phosphate export rates in the form of G1P were observed in parasites supplemented with galactose, as compared to parasites supplemented with glucose. These results suggest that an important amount of the carbon derived from galactose oxidation could be metabolized mainly through cytosolic PPP, and consequently result in higher redox buffering capabilities.

## Figures and Tables

**Figure 1 pathogens-11-01174-f001:**
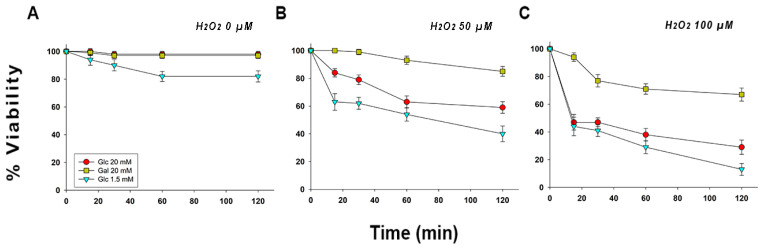
Viability of *T. cruzi* epimastigotes exposed to hydrogen peroxide with different carbon and energy sources. (**A**) Epimastigotes maintained in OSB supplemented with 20 mM glucose, 20 mM galactose or without hexoses (Glc 1.5 mM),1 and the absence of hydrogen peroxide (0 µM). (**B**) Epimastigotes maintained in OSB supplemented with 20 mM glucose, 20 mM galactose, or without hexoses (Glc 1.5 mM) in the presence of hydrogen peroxide 50 µM. (**C**) Epimastigotes maintained in OSB supplemented with 20 mM glucose, 20 mM galactose, or without hexoses (Glc 1.5 mM) in the presence of 100 µM hydrogen peroxide. The means and SD were calculated with three independent experiments.

**Figure 2 pathogens-11-01174-f002:**
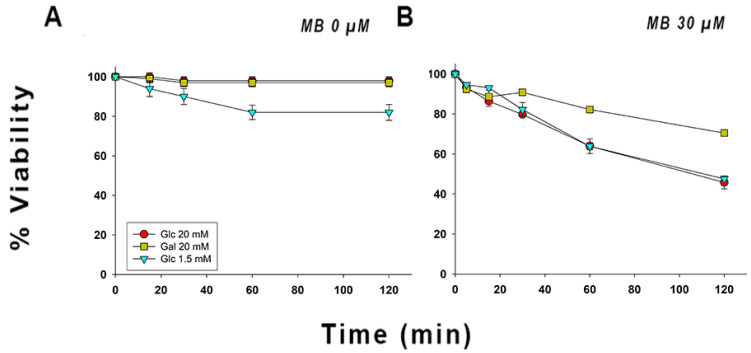
Viability percentages of *T. cruzi* epimastigotes incubated in the presence of MB and supplemented with different carbon and energy sources. (**A**) Epimastigotes maintained in OSB supplemented with the respective carbon source in the absence of MB. (**B**) Epimastigotes maintained in OSB supplemented with the respective carbon source in the presence of MB 30 µM. The means and SD were calculated with three independent experiments.

**Figure 3 pathogens-11-01174-f003:**
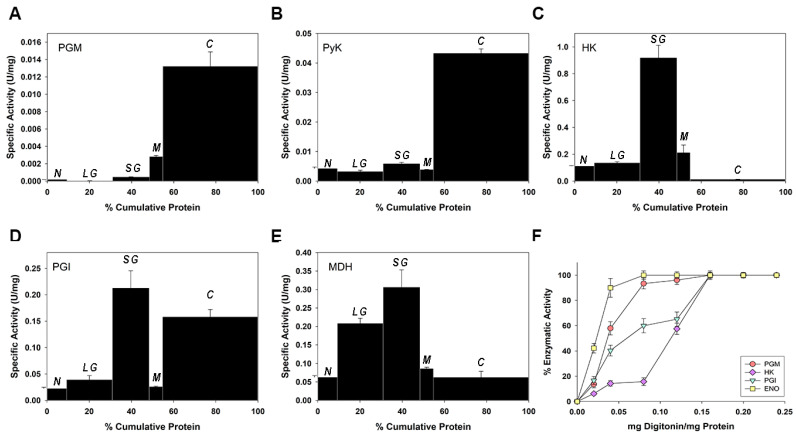
Subcellular localization of PGM activity in *Trypanosoma cruzi* epimastigotes. Panels A–E: differential centrifugation of a homogenate of *T. cruzi* epimastigotes. Fractions are plotted according to their isolation order, from left to right: N, nuclear; LG, large granular; SG, small granular; M, microsomal; and C, cytosol. (**A**) PGM, (**B**) PyR, (**C**) HK, (**D**) PGI, (**E**) MDH. Panel (**F**): partial permeabilization with digitonin of *T. cruzi* epimastigotes. The means and SD were calculated with three independent experiments.

**Figure 4 pathogens-11-01174-f004:**
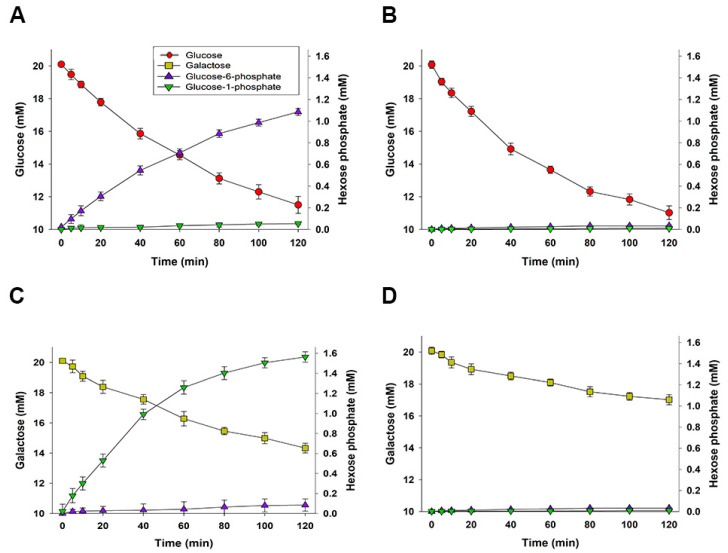
Inorganic phosphate-dependent transport of G1P and G6P in partially permeabilized *T. cruzi* epimastigotes supplemented with glucose or galactose. (**A**) Glucose consumption, G1P release, and G6P release in the presence of 6 mM inorganic phosphate. (**B**) Glucose consumption, G1P release, and G6P release in the absence of inorganic phosphate. (**C**) Galactose consumption, G1P release, and G6P release in the presence of 6 mM inorganic phosphate. (**D**) Galactose consumption, G1P release, and G6P release in the absence of inorganic phosphate. The means and SD are calculated with three independent experiments.

**Figure 5 pathogens-11-01174-f005:**
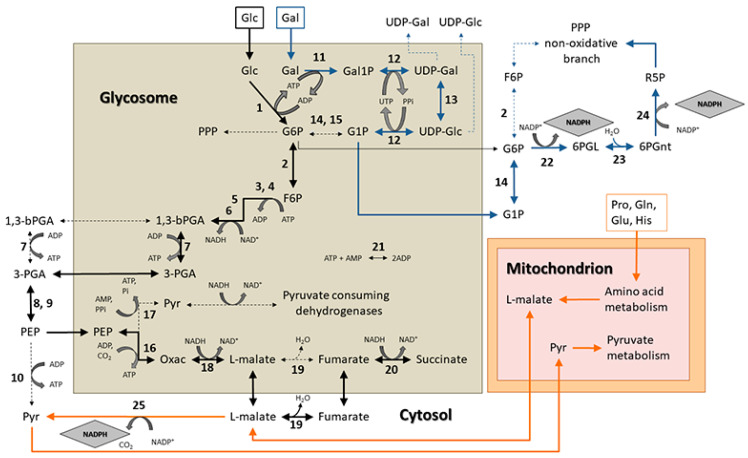
Schematic representation of interactions between glucose, galactose, and amino acid metabolism oriented toward the production of cytosolic NADPH in *Trypanosoma cruzi* epimastigotes. Black, orange, and blue arrows represent the main steps of glucose, amino acids, and galactose metabolism, respectively. Arrows with different thicknesses tentatively represent the metabolic flux of each enzymatic step. The glycosome, cytosol, and mitochondrion compartments are indicated. Cytosolic NADPH is highlighted enclosed in a rhombus. Abbreviations: 1,3 dPG, 1,3 biphosphoglycerate; 3-PGA, 3-phosphoglycerate; 6PGL, 6-phosphoglucono-γ-lactone; 6PGnt, 6-phosphogluconate; ATP/ADP/AMP, Adenosine 5′-tri, di or monophosphate; F6P, fructose 6-phosphate; G1P, glucose 1-phosphate; G6P, glucose 6-phosphate; Gal, D-galactose; Gal1P, galactose 1-phosphate; Glc, D-glucose; Gln, L-glutamine; Glu, L-glutamate; His, L-histidine; NAD(P)+/NAD(P)H, nicotinamide adenine dinucleotide (phosphate) oxidized and reduced forms; Oxac, oxaloacetate; PEP, phosphoenolpyruvate; Pi, inorganic phosphate; PPi, inorganic pyrophosphate; PPP, pentose phosphate pathway; Pro, L-proline; Pyr, pyruvate; R5P, ribulose 5-phosphate; UDP-Gal, UDP-galactose; UDP-Glc, UDP-glucose; UTP, uridine 5′-triphosphate. Enzymes: 1, hexokinase; 2, phosphoglucose isomerase; 3, phosphofructokinase; 4, aldolase; 5, triose-phosphate isomerase; 6, glyceraldehide-3-phosphate dehydrogenase; 7, phosphoglycerate kinase; 8, phosphoglycerate mutase; 9, enolase; 10, pyruvate kinase; 11, galactokinase; 12, UDP-sugar pyrophosphorylase; 13, UDP-galactose 4-epimerase; 14, phosphoglucomutase; 15, phosphomannomutase; 16, phosphoenolpyruvate carboxykinase; 17, pyruvate phosphate dikinase; 18, malate dehydrogenase; 19, fumarase; 20, fumarate dehydrogenase; 21, adenylate kinase; 22 glucose-6-phosphate dehydrogenase; 23, lactonase; 24, 6-phosphogluconate dehydrogenase, 25, malic enzyme.

**Table 1 pathogens-11-01174-t001:** Specific activities of several enzymes quantified from *T. cruzi* epimastigote homogenates grown in a LIT medium supplemented with different carbon sources.

	Specific Activity (nmol × min^−1^ × mg^−1^)		
Enzymes	Glucose 20 mM	Glucose 1.5 mM	Galactose 20 mM
Hexokinase	409.0 ± 67.0	408.0 ± 42.0	381.0 ± 24.0
Glucose-6-phosphate dehydrogenase	31.1 ± 2.8	20.1 ± 2.8	30.2 ± 3.3
Galactokinase	38.4 ± 4.3	28.7 ± 1.4	125.8 ± 11.3
Phosphoglucomutase	3.8 ± 0.1	2.5 ± 0.2	7.0 ± 0.6
NAD(H)-dependent L-glutamate dehydrogenase	179.5 ± 5.3	72.5 ± 5.9	224 ± 27.9

The means and SD are calculated with three independent experiments.

**Table 2 pathogens-11-01174-t002:** The generation time, hexose consumption rates, and ammonium excretion rates of *T. cruzi* epimastigotes grown using different carbon sources.

LIT Medium Type	Generation Time (Hours)	Hexose Consumption Rate (*q*) (nmol × min^−1^ × mg^−1^)	Ammonium Excretion Rate (*q*) (nmol × min^−1^ × mg^−1^)
Glucose 20 mM	18.79 ± 0.49	5.41 ± 0.58	2.95 ± 0.78
Glucose 1.5 mM	54.56 ± 4.95	−−−−	5.99 ± 1.14
Galactose 20 mM	20.13 ± 0.69	2.73 ± 0.58	9.72 ± 1.97

The means and SD are calculated with three independent experiments.

## Data Availability

The data presented in this study are available on request from the corresponding authors.

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
