# Peer review of "Consumption of Galactose by *Trypanosoma cruzi* Epimastigotes Generates Resistance against Oxidative Stress"

_pathogens, 2022, doi:10.3390/pathogens11101174_

Round 1

Reviewer 1 Report

The present manuscript was aimed to demonstrate that galactose usage as a primary carbon source by T. cruzi epimastigotes, provides higher resistance against oxidative stress exposure as compared to glucose. In addition, the authors study the possible adapted metabolic modifications that would be involved in this response when galactose is used as main carbon source. Thus, authors determine that phosphoglucomutase activity occurs predominantly within the T. cruzi cytosolic compartment. Furthermore, they describe that intact glycosomes from epimastigotes incubated with galactose 20 mM, released a higher glucose-1-phosphate molecules as compared to intact glycosomes from epimastigotes treated with glucose, which predominantly released glucose-6-phosphate. Both results strongly suggest that an important rate of galactose oxidation could be metabolized mainly through cytosolic from the pentose phosphate pathway, and consequently result in higher redox buffering capabilities by NAPDH producing. In my opinion, with simple and direct experiments, they obtain results that deserve to be published. However, in my opinion, there are some aspects that can be improved.

Comments:

-        Maybe the title should be shortened: “Consumption of galactose by Trypanosoma cruzi epimastigotes generates resistance against oxidative stress”

-        In the results section (page 3, lane 99), the following sentence is mentioned: “In the 50 μM of hydrogen peroxide assay, the effect of exposition begins to be evident after 120 min”. In my opinion, the viability effects are noticeable even before 120 min. Missing to add a statistical analysis to the figure.

-        In the results section (page 5, lane 203), the following sentence is mentioned: “unlike phosphoglucomutase (PGI) which is located in both the cytosol (C) and the glyco-203 some (SG)…. PGI is a phosphoglucose isomerase. Nowhere in the text is it mentioned.

-        Perhaps in figure 3 it would indicate in all the panels the order of isolation of the subcellular compartments, with the aim of making the interpretation more "friendly" to the reader. Again, missing to add the statistical analysis to the figure.

-        In the results section (page 6, lane 232), the following sentence is mentioned: “NAD(H)-dependent L-glutamate dehydrogenase (GlDH-NAD+, amino acid degradation marker) also shows a specific activity that is consistent with previous results”. Consistent with what? Values obtained with epimastigotes grown in standard LIT medium (glucose 20 mM)?. The phrase is not well understood

-        In the results section (page 7, lane 251), the following sentence is mentioned: “Parasites supplemented with galactose 20 mM showed a “similar G” when compared to parasites….”. Although what "G" means is in material and methods (Epimastigote generation Time), it would be highly desirable that its meaning be specified in the introduction or results.

-        In the text (example CO2 or H2O), correct the numbers with subscript letter

-        In the conclusion section (page 15, lane 621), the following sentence is mentioned: “These results suggest that an important amount of the carbon derived from galactose oxidation could be metabolized mainly through cytosolic PPP, and consequently result in higher redox buffering capabilities”. In my opinion, it would include that it is not ruled out that an overexpression of detoxifying systems enzymes is also generated.

Author Response

Answers to the reviewer # 1:

  • Maybe the title should be shortened: “Consumption of galactose by Trypanosoma cruziepimastigotes generates resistance against oxidative stress”

                        Answer:

The issue was addressed as the reviewer suggested, the title was shortened and compounds hydrogen peroxide and methylene blue were added as keywords.

  • In the results section (page 3, lane 99), the following sentence is mentioned: “In the 50 μM of hydrogen peroxide assay, the effect of exposition begins to be evident after 120 min”. In my opinion, the viability effects are noticeable even before 120 min. Missing to add a statistical analysis to the figure.

                        Answer:

The reviewer has suggested adding a statistical analysis to this figure, perhaps because senses that may be a significant difference among results presented, in fact the reviewer is right and a statistically significant difference was found. The statistical analysis is presented in a new table added to supplementary material.

As a consequence the sentence that is mentioned by the reviewer was changed to: “In the 50 μM of hydrogen peroxide assay, the effect of exposition begins to be evident just after initiating the experiment at 15 min of exposition”

  • In the results section (page 5, lane 203), the following sentence is mentioned: “unlike phosphoglucomutase (PGI) which is located in both the cytosol (C) and the glyco-203 some (SG)…. PGI is a phosphoglucose isomerase. Nowhere in the text is it mentioned.

Answer:

This typing error was corrected too. It was changed to: “… unlike phosphoglucose isomerase (PGI) which is located in both the cytosol (C) and the glycosome…”

  • Perhaps in figure 3 it would indicate in all the panels the order of isolation of the subcellular compartments, with the aim of making the interpretation more "friendly" to the reader. Again, missing to add the statistical analysis to the figure.

                        Answer:

The order of isolation of fractions was indicated in each panel for easy reading. Nevertheless the statistical analysis was not made for these experiments because each point in the dataset is independent. In the case of differential centrifugation there is no way of comparing specific activities of different enzymes; the focus of this figure is the comparison of the area among fractions which is correlated with the level of specific activity in each compartment. Meanwhile, in the figure of selective permeabilization with digitonin, it is used increasing concentrations of digitonin and each point in the dataset is independent; the study is based on the comparison of the profile of the study enzyme and the marker enzymes.

  • In the results section (page 6, lane 232), the following sentence is mentioned: “NAD(H)-dependent L-glutamate dehydrogenase (GlDH-NAD+, amino acid degradation marker) also shows a specific activity that is consistent with previous results”. Consistent with what? Values obtained with epimastigotes grown in standard LIT medium (glucose 20 mM)?. The phrase is not well understood

            Answer:

The phrase was edited to “NAD(H)-dependent L-glutamate dehydrogenase (GlDH-NAD+, amino acid degradation marker) also shows a specific activity that is consistent with previous results obtained in epimastigotes grown in standart LIT medium [45]–[47]. GALK also shows consistency with previous results… ”

  • In the results section (page 7, lane 251), the following sentence is mentioned: “Parasites supplemented with galactose 20 mM showed a “similar G” when compared to parasites….”. Although what "G" means is in material and methods (Epimastigote generation Time), it would be highly desirable that its meaning be specified in the introduction or results.

Answer:

This detail has been addressed as reviewer suggested. The meaning of “G” was mentioned in Results section, page 7, line 251.

  • In the text (example CO2 or H2O), correct the numbers with subscript letter

                        Answer:

The paper was carefully revised and that numbers or symbols that should be subscripted or superscripted were corrected.

  • In the conclusion section (page 15, lane 621), the following sentence is mentioned: “These results suggest that an important amount of the carbon derived from galactose oxidation could be metabolized mainly through cytosolic PPP, and consequently result in higher redox buffering capabilities”. In my opinion, it would include that it is not ruled out that an overexpression of detoxifying systems enzymes is also generated.

                        Answer:

The reviewer is right in the sense that the sentence is categorical and we devoid of experimental evidence to rule out certain possibilities, but we have to take into account that independently of whether our conditions are inducing overexpression of detoxifying system enzymes; these enzymes are catalysts whose reaction needs not just the enzyme and the ROS/RNS, but also the source of electrons in order to complete the reaction, and this source of electrons, according to our current state of knowledge, comes exclusively in last instance from NADPH. Nevertheless, we cannot rule out the possibility that our conditions (consumption of galactose) could induce the production of NADPH from aminoacids, which consumption is in fact enhanced. That is why we have revised this sentence in conclusions section and similarly along the discussion section too.

Reviewer 2 Report

I agree that the data support the conclusion the T. cruzi epimastigotes are more resistant to oxidative stress when grown in galactose as compared to glucose. However, there is a lot of speculation about the role of NADPH levels contributing to this resistance.  The authors should test this hypothesis directly by measuring the levels of these metabolites in the glycosome and cytosol.  

I think it is important to limit speculations about metabolic flux without metabolic labeling experiments, which I understand are perhaps outside the scope of this work.  Metabolic flux is more than a sum of the specific activities of enzymes in the pathway.

I would like to have a more explicit justification for the experiments in Figure 4 regarding G1P release into the cytoplasm as I am not sure how they relate to the other figures. 

Can the authors provide more explicit justification for the statements in lines 260-263 that galactose rates are too high to be sustained by only the PGM SA in the glycosome.

Can the authors provide more insight into the propose mechanism of Glucose1-phosphate out of the glycosome and the importance of this for their model.

I struggled with assessing the relevance of much of the discussion to the experiments done. Authors should either more more explicit about the connection or remove the extraneous conjecture.  For example--lines 366-389.  Perhaps a detailed description of their proposed model of galactose utilization in Figure 5 would help to tie these seemingly unrelated experiment approaches.

Author Response

Answers to the reviewer # 2:

The reviewer # 2 wrote this first paragraph: “I agree that the data support the conclusion the T. cruzi epimastigotes are more resistant to oxidative stress when grown in galactose as compared to glucose. However, there is a lot of speculation about the role of NADPH levels contributing to this resistance. The authors should test this hypothesis directly by measuring the levels of these metabolites in the glycosome and cytosol.”

Answer:

The second reviewer invites us to limit speculations about NADPH role and claims for a direct quantification of NADPH / NADP+ ratio into the two involved subcellular compartments.

We agree with the reviewer that we should limit speculation about NADPH role because we have not a direct quantification about this respect we have revised the manuscript in order to edit and diminish the NADPH-based speculations. In fact we thought how to obtain a direct evidence for sustaining our hypothesis; by either measuring NADPH/NADP+ or dihydrotrypanothione / trypanothione disulfide ratios. In our laboratory we devoid of any system to measure dihydrotrypanothione, and the possibility to buy one of the cheapest reagents as “Ellman's reagent” is not possible because its price in our country is more than 500 USD, nowadays this quantity is out of our budget capacity (In fact, we received a waiver for the pay fee from the editorial team of Pathogens).

 Conversely, although NADPH and NADP+ measurements seem to be more expensive we fortunately have in stock to carry out the measurements of NADPH or NADP+ by enzymatic methods. We tried to quantify the concentrations of NADPH and NADP+ by using enzymatic assays with cell extracts of epimastigotes without succeed, one reason of our failure may be due to inter-conversion between oxidized and reduced forms of the nicotinamide nucleotides which has been the major barrier to accurately measure the NADPH/ NADP+ ratio (Lu et al, 2018). Next, we tried other approximation as recommended by Kern et al (2014); we made a basic and acid extraction of NADP(H) and immediately we measured by enzymatic assays the NADP(H) without succeed, our new failure may be due to background optic density that precludes the accurately measure of nicotinamide nucleotides because of some molecules that have similar spectral properties (Sullivan et al, 2011). Finally, we devoid of any system that could help us to make an accurately measurement of nicotinamide nucleotides as HPLC coupled to fluorescence detection or LC-MS so we gave up on the idea of measuring directly NADP(H).

Indeed, the separately quantification of NADPH/NADP+ ratios in different compartments appears to be a challenging process; since any disturbance of the cells that let to obtain subcellular fractionation lead to inter-conversion between oxidized and reduced forms, and the pools usually are mixed after such treatments. To address this issue, some researchers have employed heterologous overexpression of sensor enzymes in order to measure in vivo these important cofactors in the cytosol of Saccharomyces cerevisiae (Zhang et al, 2015). Such kind of experiment is out of the scope of our research.

That is why we preferred another approximation: measuring the hexoses phosphate (glucose-6-phosphate and glucose-1-phosphate) that exit the glycosomes in partially permeabilized-epimastigotes with digitonin by using parasites that were fed with either glucose or galactose. Despite the result presented is not a direct evidence of higher redox buffering capacity of epimastigotes, it is strong evidence that agrees with our hypothesis because the only know pathway that can feed the glucose-6-phosphate and glucose-1-phosphate in the cytosol of Trypanosoma cruzi epimastigotes is the pentose phosphate pathway, as a probable consequence, produces reduce power in the form of NADPH. Based on the results obtained in this experiment we consider is enough to fulfill the expectative of Pathogens audience.

            Second paragraph: “I think it is important to limit speculations about metabolic flux without metabolic labeling experiments, which I understand are perhaps outside the scope of this work.  Metabolic flux is more than a sum of the specific activities of enzymes in the pathway.”

                        Answer:

We agree and we have met this issue by limiting speculations about metabolic flux in the revised manuscript.

            Third paragraph: “I would like to have a more explicit justification for the experiments in Figure 4 regarding G1P release into the cytoplasm as I am not sure how they relate to the other figures.”

                        Answer:

We welcome these kinds of suggestions because they help to broaden the audience of the research as it becomes much more explicit and accessible. This experiment is key for understanding the connection between Isselbacher pathway (from galactose to glucose-1-phosphate), and the possible mechanism of action in order to confer resistance against oxidative stress, which needs the action of phosphoglucomutase enzyme that occurs mainly in the cytosol compartment. As a consequence, the anti-porting mechanism of hexoses phosphate that exits the glycosome was studied. Once glucose-1-P is found into cytosol, could serve as substrate of phosphoglucomutase to produce glucose-6-phosphate which feeds the unique known glucose-6-phosphate-consuming pathway in the cytosol of these organisms: the pentose phosphate pathway. The pentose phosphate pathway oxidative branch produces NADPH which could serve as the source of electrons necessary for conferring resistance against oxidative stress.

            Fourth paragraph: “Can the authors provide more explicit justification for the statements in lines 260-263 that galactose rates are too high to be sustained by only the PGM SA in the glycosome.”

                        Answer:

Specific activity in cell extracts of PGM was estimated in 7 ± 0.6 nmol × min-1 × mg-1 (Table 1), but taking into account digitonin fractionation experiment, ~10 % of this activity is associated with glycosome compartment (figure 3); as a consequence, 0.7 nmol × min-1 × mg-1 is by far much less than the galactose consumption rate measured of 2.73 nmol × min-1 × mg-1 (Table 2), which means that galactose consumption rate is ~4-fold than PGM activity inside glycosome, that is why we looked for one way of diverting the metabolic flux to cytosol compartment where the main part of PGM activity resides. Of course, we agree that this statement cannot be made because we devoid of metabolic flux data and metabolic interactions are much more than the sum of the specific activities of the involved enzymes. For instance, glycosomal PGM could be in presence of an unknown activator, and/or a non-depreciable part of metabolic flux may be diverting to NDP-hexoses synthesis for glycoconjugates biosynthesis. That is why we have decided to edit these statements in order to avoid speculations about metabolic flux.

Fifth paragraph: “Can the authors provide more insight into the propose mechanism of Glucose1-phosphate out of the glycosome and the importance of this for their model.”

Answer:

The suggestion of the reviewer has been assessed; we have added some lines in both, the results and discussion sections in order to make more understandable the importance of exporting glucose-1-phosphate to the cytosol.

Sixth paragraph: “I struggled with assessing the relevance of much of the discussion to the experiments done. Authors should either more more explicit about the connection or remove the extraneous conjecture.  For example--lines 366-389.  Perhaps a detailed description of their proposed model of galactose utilization in Figure 5 would help to tie these seemingly unrelated experiment approaches.

Answer:

We understand that maybe discussion devoid of a phenomenological arrange that is correlated with the logic order of experiments. In fact, the aim of the discussion was to weave in a transversal mode the different aspects, for instance, physiological, host-parasite interactions, immune evasion strategies, life cycle, etc. that are deeply correlated. The discussion was formatted in that way, mainly because in results section we discuss part of the results from a phenomenological perspective. But taking into account the reviewer suggestion, we have strategically added references to the model shown in figure 5 along the discussion to better exploit the capability of this figure of transferring the main idea of the research presented.

 References:

Lu W, Wang L, Chen L, Hui S, Rabinowitz JD. Extraction and Quantitation of Nicotinamide Adenine Dinucleotide Redox Cofactors. Antioxid Redox Signal. 2018 Jan 20;28(3):167-179. doi: 10.1089/ars.2017.7014. Epub 2017 Jul 19. PMID: 28497978; PMCID: PMC5737638.

Kern SE, Price-Whelan A, Newman DK. Extraction and measurement of NAD(P)(+) and NAD(P)H. Methods Mol Biol. 2014;1149:311-23. doi: 10.1007/978-1-4939-0473-0_26. PMID: 24818916.

Sullivan NL, Tzeranis DS, Wang Y, So PT, Newman D. Quantifying the dynamics of bacterial secondary metabolites by spectral multiphoton microscopy. ACS Chem Biol. 2011 Sep 16;6(9):893-9. doi: 10.1021/cb200094w. Epub 2011 Jul 15. PMID: 21671613; PMCID: PMC3212935.

Zhang J, ten Pierick A, van Rossum HM, Seifar RM, Ras C, Daran JM, Heijnen JJ, Wahl SA. Determination of the Cytosolic NADPH/NADP Ratio in Saccharomyces cerevisiae using Shikimate Dehydrogenase as Sensor Reaction. Sci Rep. 2015 Aug 5;5:12846. doi: 10.1038/srep12846. PMID: 26243542; PMCID: PMC4525286.

Round 2

Reviewer 2 Report

I appreciate the time and effort the authors have taken to address my concerns and am feel that have addressed them appropriately.